# GAUSSIAN PROCESS NEURONS

## ABSTRACT

We propose a method to learn stochastic activation functions for use in probabilistic neural networks. First, we develop a framework to embed stochastic activation functions based on Gaussian processes in probabilistic neural networks. Second, we analytically derive expressions for the propagation of means and covariances in such a network, thus allowing for an efficient implementation and training without the need for sampling. Third, we show how to apply variational Bayesian inference to regularize and efficiently train this model. The resulting model can deal with uncertain inputs and implicitly provides an estimate of the confidence of its predictions. Like a conventional neural network it can scale to datasets of arbitrary size and be extended with convolutional and recurrent connections, if desired.

## 1 INTRODUCTION

The popularity of deep learning and the implied race for better accuracy and performance has lead to new research of the fundamentals of neural networks. Finding an optimal architecture often focusses on a hyperparameter search over the network architecture, regularization parameters, and one of a few standard activation functions: tanh, ReLU (Glorot et al. (2011)), maxout (Goodfellow et al. (2013)) ... Focussing on the latter, looking into activation functions has only taken off since Nair & Hinton (2010) introduced the rectified linear unit (ReLU), which were shown to produce significantly better results on image recognition tasks Krizhevsky et al. (2012). Maas et al. (2013) then introduced the leaky ReLU, which has a very small, but non-zero, slope for negative values. He et al. (2015) proposed the parameterized ReLU, by making the slope of the negative part of the leaky ReLU adaptable. It was trained as an additional parameter for each neuron alongside the weights of the neural network using stochastic gradient descent. Thus, the activation function was not treated as a fixed hyper-parameter anymore but as adaptable to training data. While the parameterized ReLU only has one parameter, this was generalized in Agostinelli et al. (2014a) to piecewise linear activation functions that can have an arbitrary (but fixed) number of points where the function changes it slope. This can be interpreted as a different parameterization of a Maxout network (Goodfellow et al. (2013)), in which each neuron takes the maximum over a set of different linear combinations of its inputs.

Instead of having a fixed parameter for the negative slope of the ReLU, Xu et al. (2015) introduced stochasticity into the activation function by sampling the value for the slope with each training iteration from a fixed uniform distribution. Clevert et al. (2015) and Klambauer et al. (2017) replaced the negative part of ReLUs with a scaled exponential function and showed that, under certain conditions, this leads to automatic renormalization of the inputs to the following layer and thereby simplifies the training of the neural networks, leading to an improvement in accuracy in various tasks.

Nearly fully adaptable activation functions have been proposed by Eisenach et al. (2016). The authors use a Fourier basis expansion to represent the activation function; thus with enough coefficients any (periodic) activation function can be represented. The coefficients of this expansion are trained as network parameters using stochastic gradient descent or extensions thereof.

Promoting a more general approach, Agostinelli et al. (2014b) proposed to learn the activation functions alongside the layer weights. Their adaptive piecewise linear units consist of a sum of hinge-shaped functions with parameters to control the hinges and the slopes of the linear segments. However, by construction the derivative of these activation functions is not continuous at the joints between two linear segments, which often leads to non-optimal optimizer performance.

To our knowledge, previous research on learning activation functions took place in a fully deterministic setting, i.e. deterministic activation functions were parameterized and included in the optimization of a conventional neural network. Here instead, we explore the setting of probabilistic activation functions embedded in a graphical model of random variables resembling the structure of a neural network. We develop the theory of Gaussian-process neurons and subsequently derive a lower-bound approximation using variational inference, in order to develop a computationally efficient version of the Gaussian Process neuron.

NOTATION

To define the model we will need to slice matrices along rows and columns. Given a matrix $X$, we will write $X_{i\star}$ to select all elements of the $i$-th row and $X_{\star j}$ to select all elements of the $j$-th column.

## 2 GAUSSIAN PROCESSES

Gaussian Processes (GPs) are nonparametric models that provide flexible probabilistic approaches for function estimation. A Gaussian Process (Rasmussen & Williams, 2006) defines a distribution over a function $f(x) \sim \mathcal{GP}(m(x), k(x, x'))$ where $m$ is called the mean function and $k$ is the covariance function. For $S$ inputs $\boldsymbol{x} \in \mathbb{R}^{S \times N}$ of $N$ dimensions the corresponding function values $\boldsymbol{f}$ with $f_i \triangleq f(X_{i\star})$ follow a multivariate normal distribution[1]

$$\boldsymbol{f} \sim \mathcal{N}(\boldsymbol{m}, K(X, X))$$

with mean vector $m_i \triangleq m(X_{i\star})$ and $K(X, X)$ is the covariance matrix defined by $(K(X, X))_{ij} \triangleq k(X_{i\star}, X_{j\star})$. In this work we use the zero mean function $m(x) = 0$ and the squared exponential (SE) covariance function with scalar inputs,

$$k(x, x') = \exp\left(-\frac{(x - x')^2}{2\,\nu^2}\right),\tag{1}$$

where $\nu$ is called the length-scale and determines how similar function values of nearby inputs are according to the GP distribution. Since this covariance function is infinitely differentiable, all function samples from a GP using it are smooth functions.

## 3 GAUSSIAN PROCESS NEURONS

We will first describe the fundamental, non-parametric model, which will be approximated in the following sections for efficient training and inference. Let the input to the $l$-th layer of Gaussian Process neurons (GPNs) be denoted by $X^{l-1} \in \mathbb{R}^{S \times N_{l-1}}$ where $S$ is the number of data points (samples) and $N_{l-1}$ is the number of input dimensions. A layer $l \in \{1, \ldots, L\}$ of $N_l$ GPNs indexed by $n \in \{1, \ldots N_l\}$ is defined by the joint probability

$$\mathrm{P}(X^l_{\star n}, F^l_{\star n} \,|\, X^{l-1}) = \mathrm{P}(F^l_{\star n} \,|\, X^{l-1})\,\mathrm{P}(X^l_{\star n} \,|\, F^l_{\star n})\tag{2}$$

with the GP prior $F^l$ conditioned on the layer inputs $X^{l-1}$ multiplied with the weights $W^l$,

$$F^l_{\star n} \,|\, X^{l-1} \sim \mathcal{N}(\boldsymbol{0}, K^l(X^{l-1} W^l_{\star n}, X^{l-1} W^l_{\star n})),\tag{3}$$

and an additive Gaussian noise distribution,

$$X^l_{\star n} \,|\, F^l_{\star n} \sim \mathcal{N}(F^l_{\star n}, (\sigma^l_n)^2 \mathbb{1}).\tag{4}$$

This corresponds to a probabilistic activation function $F^l_{sn} = f^l_n(X^{l-1}_{s\star} W^l_{\star n})$ with

$$f^l_n(z) \sim \mathcal{GP}(0, k(z, z')).\tag{5}$$

This GP has *scalar* inputs and uses the standard squared exponential covariance function. Analogous to standard neural networks, GPN layers can be stacked to form a multi-layer feed-forward network. The joint probability of such a stack is

$$\mathrm{P}(X^{\{l\}}, F^{\{l\}}) = \mathrm{P}(X^0) \prod_{l=1}^{L} \mathrm{P}(X^l_{\star n}, F^l_{\star n} \,|\, X^{l-1}).\tag{6}$$

---

[1]We omit writing $P(\bullet)$.

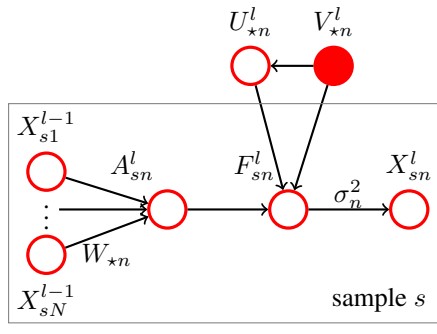

Figure 1: The auxiliary parametric representation of a GPN using virtual observation inducing points $V_\star$ and targets $U_\star$.

All input samples $X_{s\star}^0$, $s \in \{1, \dots, S\}$, are assumed to be normally distributed with known mean and covariance,

$$\mathrm{P}(X^0) = \prod_{s=1}^{S} \mathcal{N}(X_{s\star}^0 \,|\, \mu_{s\star}^{\mathrm{in}}, \Sigma_{s\star\star}^{\mathrm{in}}) \,. \tag{7}$$

To obtain predictions $\mathrm{P}(X^L \,|\, X^0)$, all latent variables in eq. (6) would need to be marginalized out; unfortunately due to the occurrence of $X^l$ in the covariance matrix in eq. (3) analytic integration is intractable.[2]

## 3.1 AUXILIARY PARAMETRIC REPRESENTATION

The path to obtain a tractable training objective is to temporarily parameterize the activation function $f_n^l(z)$ of each GPN using virtual observations (originally proposed by Quiñonero-Candela & Rasmussen (2005) for sparse approximations of GPs) of inputs and outputs of the function. These virtual observations are only introduced as an auxiliary device and will be marginalized out later. Each virtual observation $r$ consists of a *scalar* inducing point $V_{rn}^l$ and corresponding target $U_{rn}^l$. Under these assumptions on $f_d^l$, the GP prior $\mathrm{P}(F_{\star n}^l \,|\, X^{l-1})$ is replaced with

$$F_{\star d}^l \,|\, X^{l-1}, U_{\star n}^l \sim \mathcal{N}(\mu_{\star n}^{F^l}, \Sigma_{\star\star n}^{F^l}) \tag{8}$$

where the mean and variance are those obtained by using the virtual observations as "training" points for a GP regression evaluated at the "test" points $X^l W_{\star d}^l$,

$$\mu_{\star n}^{F^l} = K(X^{l-1} W_{\star n}^l, V_{\star n}^l) \, (\widetilde{K}_{n\star\star}^l)^{-1} \, U_{\star n}^l \tag{9}$$

$$\Sigma_{\star\star d}^{F^l} = K(X^{l-1} W_{\star n}^l, X^{l-1} W_{\star n}^l) - K(X^{l-1} W_{\star n}^l, V_{\star n}^l) \, (\widetilde{K}_{n\star\star}^l)^{-1} \, K(V_{\star n}^l, X^{l-1} W_{\star n}^l) \,. \tag{10}$$

where $\widetilde{K}_{nrt}^l = K(V_{rn}^l, V_{tn}^l)$.

Given enough inducing points that lie densely between the layer's activations $A_{\star n}^l = X^{l-1} W_{\star n}^l$, the shape of the activation function becomes predominantly determined by the corresponding targets of these inducing points. Consequently, the inter-sample correlation in eq. (8) becomes negligible, allowing us to further approximate this conditional by factorizing it over the samples; thus we have $\mathrm{P}(F_{\star n}^l \,|\, X^l, U_{\star n}^l) = \prod_s \mathrm{P}(F_{sn}^l \,|\, X_{s\star}^{l-1}, U_{\star n}^l)$ with

$$F_{sn}^l \,|\, X_{s\star}^{l-1}, U_{\star n}^l \sim \mathcal{N}(\mu_{sn}^{F^l}, \Sigma_{ssn}^{F^l}) \,. \tag{11}$$

We now marginalize eq. (4) over $F_{\star n}^l$ and get $\mathrm{P}(X_{\star n}^l \,|\, X^{l-1}, U_{\star n}^l) = \prod_s \mathrm{P}(X_{sn}^l \,|\, X_{s\star}^{l-1}, U_{\star n}^l)$ with

$$X_{sn}^l \,|\, X_{s\star}^{l-1}, U_{\star n}^l \sim \mathcal{N}(\mu_{sn}^{F^l}, \Sigma_{ssn}^{F^l} + (\sigma_n^l)^2) \,. \tag{12}$$

---

[2] The inverse of the covariance matrix appears in the PDF of a normal distribution. Thus, the dependency on $X^l$ is highly non-linear and analytic calculation of the integral is not feasible.

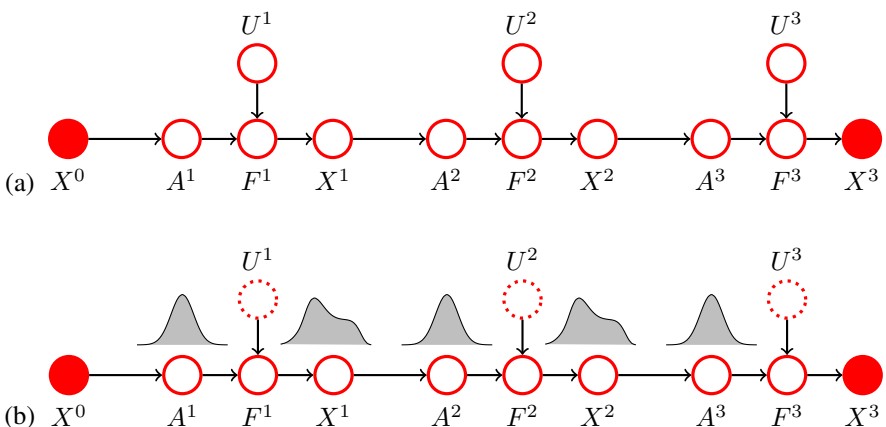

Figure 2: A GPN feed-forward network prior distribution (a) for three layers and the approximation of its posterior obtained using variational inference and the central limit theorem on the activations (b). Each node corresponds to all samples and GPN units within a layer. Dotted circles represent variational parameters.

Although we now have a distribution for $X^l$ that is conditionally normal given the values of the previous layer, the marginals $\mathrm{P}(X_{\star n}^l)$, $l \in \{1, \dots, L\}$, will, in general, not be normally distributed, because the input from the previous layer $X^{l-1}$ appears non-linearly through the kernel function in the mean eq. (9).

By putting a GP prior on the distribution of the virtual observation targets $U^l$ as shown in fig. 1.,

$$U_{\star n}^l \sim \mathcal{N}(\mathbf{0}, K(V_{\star n}^l, V_{\star n}^l)),\tag{13}$$

it can easily be verified that the marginal distribution of the response,

$$\mathrm{P}(F_{\star n}^l \mid X^l) = \int \mathrm{P}(F_{\star n}^l \mid X^l, U_{\star n}^l)\, \mathrm{P}(U_{\star n}^l)\, \mathrm{d}U_{\star n}^l = \mathcal{N}(F_{\star n}^l \mid 0, K(X^{l-1}W_{\star n}^l, X^{l-1}W_{\star n}^l)),\tag{14}$$

recovers the original, non-parametric GPN response distribution given by (3). The first use of this prior-restoring technique was presented in Titsias (2009) for finding inducing points of sparse GP regression using variational methods.

## 3.2 APPROXIMATE BAYESIAN INFERENCE FOR GPN NETWORKS

The joint distribution of a GPN feed-forward network is given by

$$\mathrm{P}(\{X\}_1^L, \{A\}_1^L, \{U\}_1^L, \{F\}_1^L \mid X^0) = \prod_{l=1}^L \mathrm{P}(A^l \mid X^{l-1})\, \mathrm{P}(U^l)\, \mathrm{P}(F^l \mid A^l, U^l)\, \mathrm{P}(X^l \mid F^l),\tag{15}$$

where to notation $\{\bullet\}_1^L$ should be read as $\{\bullet^1, \bullet^2, \dots, \bullet^L\}$. A graphical model corresponding to that distribution for three layers is shown in fig. 2a. Since exact marginalization over the latent variables is infeasible we apply the technique of variational inference Wainwright et al. (2008) to approximate the posterior of the model given some training data by a variational distribution $Q$.

The information about the activation functions learned from the training data is mediated via the virtual observation targets $U^l$, thus their variational posterior must be adaptable in order to store that information. Hence, we choose a normal distribution factorized over the GPN units within a layer with free mean and covariance for the approximative posterior of $U^l$,

$$Q(U^l) = \prod_{n=1}^{N_l} Q(U_{\star n}^l), \qquad Q(U_{\star n}^l) = \mathcal{N}(U_{\star n}^l \mid \widehat{\mu}_{\star n}^{U^l}, \widehat{\Sigma}_{\star \star n}^{U^l}).\tag{16}$$

This allows the inducing targets of a GPN to be correlated, but the covariance matrix can be constrained to be diagonal, if it is desired to reduce the number of variational parameters. We keep the

rest of the model distribution unchanged from the prior; thus the overall approximating posterior is

$$Q(\{U\}_1^L, \{X\}_1^{L-1}, \{A\}_1^L, \{F\}_1^L) = \prod_{l=1}^L P(A^l \mid X^{l-1}) \, Q(U^l) \, P(F^l \mid A^l, U^l) \, P(X^l \mid F^l) \,. \quad (17)$$

Estimating the variational parameters $\widehat{\mu}^{U^l}$ and $\widehat{\Sigma}^{U^l}$ requires maximizing the evidence lower bound (ELBO) given by

$$Ł = - \int \cdots \int Q(\{U\}_1^L, \{X\}_1^{L-1}, \{A\}_1^L, \{F\}_1^L) \log \frac{Q(\{U\}_1^L, \{X\}_1^{L-1}, \{A\}_1^L, \{F\}_1^L)}{P(\{U\}_1^L, \{X\}_1^L, \{A\}_1^L, \{F\}_1^L \mid X^0)} \,.$$
$$\mathrm{d}\{U\}_1^L \, \mathrm{d}\{X\}_1^L \, \mathrm{d}\{A\}_1^L \, \mathrm{d}\{F\}_1^L \,. \quad (18)$$

Substituting the distributions into this equation results in $Ł = -Ł_{\mathrm{reg}} + Ł_{\mathrm{pred}}$ with

$$Ł_{\mathrm{reg}} = \sum_{l=1}^L \int Q(U^l) \log \frac{Q(U^l)}{P(U^l)} \, \mathrm{d}U^l \,, \quad (19)$$

$$Ł_{\mathrm{pred}} = \int Q(F^L) \log P(X^L \mid F^L) \, \mathrm{d}F^L \,. \quad (20)$$

The term $Ł_{\mathrm{reg}}$ can be identified as the sum of the KL-divergences between the GP prior on the virtual observation targets and their approximative posterior $Q(U^l)$. Since this term enters $Ł$ with a negative sign, its purpose is to keep the approximative posterior close to the prior; thus it can be understood as a regularization term. Evaluating using the formula for the KL-divergence between two normal distributions gives

$$Ł_{\mathrm{reg}} = \sum_{l=1}^L \mathrm{KL}\big(Q(U^l) \,\|\, P(U^l)\big) = \sum_{l=1}^L \sum_{n=1}^{N_l} \mathrm{KL}\big(Q(U_{\star n}^l) \,\|\, P(U_{\star n}^l)\big) \quad (21)$$

$$\propto \frac{1}{2} \sum_{l=1}^L \sum_{n=1}^{N_l} \left( \mathrm{tr}\Big(K(V_{\star n}^l, V_{\star n}^l)^{-1} \widehat{\Sigma}_{\star\star n}^{U^l}\Big) + (\widehat{\mu}_{\star n}^{U^l})^T K(V_{\star n}^l, V_{\star n}^l)^{-1} \widehat{\mu}_{\star n}^{U^l} + \log \frac{\big|K(V_{\star n}^l, V_{\star n}^l)\big|}{\big|\widehat{\Sigma}_{\star\star n}^{U^l}\big|} \right) \,.$$

The term $Ł_{\mathrm{pred}}$ cannot be evaluated yet because the exact marginal $Q(F^L)$ is still intractable.

### 3.2.1 CENTRAL LIMIT DISTRIBUTION OF ACTIVATIONS

Due to the central limit theorem the activation $A^l$ (weighted sum of inputs) of each GPN will converge to a normal distribution, *if the number of incoming connections is sufficiently large ($\geq 50$) and* the weights $W^l$ have a sufficiently random distribution. For standard feed forward neural networks Wang & Manning (2013) experimentally showed that even *after* training the weights are sufficiently random for this assumption to hold. Hence we postulate that the same is true for GPNs and assume that the *marginal* distributions $Q(A^l)$, $l \in \{1, \dots, L\}$, can be written as

$$Q(A^l) = \prod_{s=1}^S Q(A_{s\star}^l), \qquad Q(A_{s\star}^l) \approx \mathcal{N}\left(A_{s\star}^l \,\Big|\, \mu_{s\star}^{A^l}, \Sigma_{s\star\star}^{A^l}\right) \,. \quad (22)$$

A graphical model corresponding to this approximate posterior is shown in fig. 2b. This allows the moments of $Q(A^l)$ to be calculated exactly and propagated analytically from layer to layer. For this purpose we need to evaluate the conditional distributions

$$Q(X^l \mid X^{l-1}) = \int Q(F^l \mid A^l = W^l X^{l-1}) \, P(X^l \mid F^l) \, \mathrm{d}F^l \,, \quad (23)$$

$$Q(F^l \mid A^l) = \prod_{n=1}^{N_l} \int Q(U_{\star n}^l) \, P(F_{\star n}^l \mid A_{\star n}^l, U_{\star n}^l) \, \mathrm{d}U_{\star n}^l \,. \quad (24)$$

Since $Q(F^l \mid A^l)$ is the conditional of a GP with normally distributed observations, the joint distribution $Q(F_{\star n}^l, U_{\star n}^l \mid A_{\star n}^l) = Q(U_{\star n}^l) \, P(F_{\star n}^l \mid A_{\star n}^l, U_{\star n}^l)$ must itself be normal,

$$Q(F_{\star n}^l, U_{\star n}^l \mid A_{\star n}^l) = \mathcal{N}\left( \begin{bmatrix} F_{\star n}^l \\ U_{\star n}^l \end{bmatrix} \,\Bigg|\, \begin{bmatrix} \widehat{\mu}_{\star n}^{F^l} \\ \widehat{\mu}_{\star n}^{U^l} \end{bmatrix}, \begin{bmatrix} \widehat{\Sigma}_{\star\star n}^{F^l} & \widetilde{\Sigma}_{FU} \\ \widetilde{\Sigma}_{UF} & \widehat{\Sigma}_{\star\star n}^{U^l} \end{bmatrix} \right) \,, \quad (25)$$

and we can find the values for the unknown parameters $\widehat{\mu}_{\star n}^{F^l}$, $\widehat{\Sigma}_{\star\star n}^{F^l}$ (and $\widetilde{\Sigma}_{FU} = \widetilde{\Sigma}_{UF}^T$) by equating the moments of its conditional distribution $Q(F_{\star n}^l \,|\, U_{\star n}^l, A_{\star n}^l)$ with $P(F_{\star n}^l \,|\, U_{\star n}^l, A_{\star n}^l)$. Thus by solving the resulting equations we obtain for eq. (24),

$$Q(F^l \,|\, A^l) = \prod_{n=1}^{N^l} \mathcal{N}(F_{\star n}^l \,|\, \widehat{\mu}_{\star n}^{F^l}, \widehat{\Sigma}_{\star\star n}^{F^l}) \,. \tag{26}$$

where

$$\widehat{\mu}_{\star n}^{F^l} = K(A_{\star n}^l, V_{\star n}^l)\, K(V_{\star n}^l, V_{\star n}^l)^{-1}\, \widehat{\mu}_{\star n}^{U^l} \tag{27}$$

$$\widehat{\Sigma}_{\star\star n}^{F^l} = K(A_{\star n}^l, A_{\star n}^l) - K(A_{\star n}^l, V_{\star n}^l)\, \widehat{K}_{\star\star n}^{U^l}\, K(V_{\star n}^l, A_{\star n}^l) \tag{28}$$

with

$$\widehat{K}_{\star\star n}^{U^l} \triangleq K(V_{\star n}^l, V_{\star n}^l)^{-1} - K(V_{\star n}^l, V_{\star n}^l)^{-1}\, \widehat{\Sigma}_{\star\star n}^{U^l}\, K(V_{\star n}^l, V_{\star n}^l)^{-1} \,. \tag{29}$$

For deterministic observations, that is $\widehat{\Sigma}_{\star\star n}^{U^l} = 0$, we obtain $\widehat{K}_{\star\star n}^{U^l} = K(V_{\star n}^l, V_{\star n}^l)^{-1}$ and thus recover the standard GP regression distribution as expected. If $U^l$ follows its prior, that is $\widehat{\mu}_{\star n}^{U^l} = \mathbf{0}$ and $\widehat{\Sigma}_{\star\star n}^{U^l} = K(V_{\star n}^l, V_{\star n}^l)$, we obtain $\widehat{K}_{\star\star n}^{U^l} = 0$ and thus recover the GP prior on $F^l$. In that case the virtual observations behave as if they were not present.

Having $Q(F^l \,|\, A^l)$ immediately allows us to evaluate eq. (23) since $P(X^l \,|\, F^l)$ just provides additive Gaussian noise; thus we obtain

$$Q(X^l \,|\, X^{l-1}) = \prod_{n=1}^{N^l} \mathcal{N}(F_{\star n}^l \,|\, \widehat{\mu}_{\star n}^{F^l}, \widehat{\Sigma}_{\star\star n}^{F^l} + (\sigma_n^l)^2 \mathbb{1}) \,. \tag{30}$$

Returning to $\text{Ł}_{\text{pred}}$ from (20) and writing it as

$$\text{Ł}_{\text{pred}} = \iiint Q(A^L)\, Q(U^L)\, P(F^L \,|\, A^L, U^L)\, \log P(X^L \,|\, F^L)\, \mathrm{d}A^L\, \mathrm{d}U^L\, \mathrm{d}F^L \,. \tag{31}$$

shows that we first need to obtain the distribution $Q(A^L)$. This is done by iteratively calculating the marginals $Q(A^l)$ for $l \in \{1, \dots, L\}$.

### 3.2.2 Propagation of Uncertainty

For $l \geq 1$ the marginal distribution of the activations is

$$Q(A^{l+1}) = \iiint Q(A^l)\, Q(F^l \,|\, A^l)\, P(X^l \,|\, F^l)\, P(A^{l+1} \,|\, X^l)\, \mathrm{d}A^l\, \mathrm{d}F^l\, \mathrm{d}X^l \tag{32}$$

where $Q(F^l \,|\, A^l)$ is given by (26). We first evaluate the mean and covariance of the marginal $Q(F^l) = \int Q(A^l)\, Q(F^l \,|\, A^l)\, \mathrm{d}A^l$. For the marginal mean of the response $\widetilde{\mu}_{sn}^{F^l} \triangleq \mathrm{E}_{Q(F^l)}\big[F_{sn}^l\big] = \mathrm{E}_{Q(A_{s\star}^l)}\big[\widehat{\mu}_{sn}^{F^l}\big]$ we obtain

$$\widetilde{\mu}_{sn}^{F^l} = \mathrm{E}_{Q(A_{s\star}^l)}\big[K(A_{sn}^l, V_{\star n}^l)\big]^T K(V_{\star n}^l, V_{\star n}^l)^{-1}\, \widehat{\mu}_{\star n}^{U^l} = (\psi_{s\star n}^l)^T K(V_{\star n}^l, V_{\star n}^l)^{-1}\, \widehat{\mu}_{\star n}^{U^l} \tag{33}$$

with

$$\psi_{stn}^l \triangleq \mathrm{E}_{Q(A_{sn}^l)}[K(A_{sn}^l, V_{tn}^l)] = \sqrt{\frac{\nu^2}{\nu^2 + \Sigma_{snn}^{A^l}}} \exp\left(-\frac{(\mu_{sn}^{A^l} - V_{tn}^l)^2}{2(\nu^2 + \Sigma_{snn}^{A^l})}\right) \,,$$

which was calculated by expressing the squared exponential kernel as a normal PDF and applying the product formula for Gaussian PDFs (Bromiley, 2003). For the marginal covariances of the response $\widetilde{\Sigma}_{s\star\star}^{F^l}$ we obtain by applying the law of total expectation

$$\widetilde{\Sigma}_{snn'}^{F^l} \triangleq \mathrm{Cov}(F_{sn}^l, F_{sn'}^l) = \mathrm{E}_{Q(A_{s\star}^l)}\Big[\mathrm{E}_{Q(F_{sn}^l, F_{sn'}^l \,|\, A^l)}[F_{sn}^l\, F_{sn'}^l]\Big] - \widetilde{\mu}_{sn}^{F^l}\, \widetilde{\mu}_{sn'}^{F^l} \tag{34}$$

For the elements representing the variance, i.e. the diagonal $n = n'$, this becomes

$$\widetilde{\Sigma}_{snn}^{F^l} = \mathrm{E}_{Q(A_{s\star}^l)}\Big[\widehat{\Sigma}_{ssn}^{F^l} + (\widehat{\mu}_{sn}^{F^l})^2\Big] - (\widetilde{\mu}_{sn}^{F^l})^2$$

$$= 1 - \mathrm{Tr}\Big([\widehat{K}_{\star\star n}^{U^l} - \beta_{\star n}^l(\beta_{\star n}^l)^T]\, \Omega_{s\star\star n}^l\Big) - \mathrm{Tr}\big(\psi_{s\star n}^l(\psi_{s\star n}^l)^T\, \beta_{\star n}^l(\beta_{\star n}^l)^T\big) \tag{35}$$

with $\beta_{\star n}^l \triangleq K(V_{\star n}^l, V_{\star n}^l)^{-1} \widehat{\mu}_{\star n}^{U^l}$ and, using the same method as above,

$$\Omega_{strn}^l \triangleq \mathrm{E}_{\mathrm{Q}(A_{s\star}^l)}[K(A_{sn}^l, V_{tn}^l)\, K(A_{sn}^l, V_{rn}^l)]$$

$$= \sqrt{\frac{\nu^2}{\nu^2 + 2\Sigma_{snn}^{A^l}}} \exp\left( -\frac{\left(\mu_{sn}^{A^l} - \frac{V_{tn}^l + V_{rn}^l}{2}\right)^2}{\nu^2 + 2\Sigma_{snn}^{A^l}} - \frac{(V_{tn}^l - V_{rn}^l)^2}{4\nu^2} \right). \tag{36}$$

For off-diagonal elements, $n \neq n'$, we observe that $F_{sn}^l$ and $F_{sn'}^l$ are *conditionally independent* given $A^l$ because the activation functions of GPNs $n$ and $n'$ are represented by two different GPs. Hence we have

$$\widetilde{\Sigma}_{snn'}^{F^l} = \mathrm{E}_{\mathrm{Q}(A_{s\star}^l)}\left[ \widehat{\mu}_{sn}^{F^l} \widehat{\mu}_{sn'}^{F^l} \right] - \widetilde{\mu}_{sn}^{F^l} \widetilde{\mu}_{sn'}^{F^l} = (\beta_{\star n}^l)^T \Lambda_{s\star\star nn'}^l \beta_{\star n'}^l - \widetilde{\mu}_{sn}^{F^l} \widetilde{\mu}_{sn'}^{F^l}. \tag{37}$$

where

$$\Lambda_{srtnn'}^l \triangleq \mathrm{E}_{\mathrm{Q}(A_{s\star}^l)}\left[ K(A_{sn}^l, V_{tn}^l)\, K(A_{sn'}^l, V_{rn'}^l) \right] = \frac{\nu^2 \exp(\mathscr{A}_{srtnn'}^l / \mathscr{B}_{snn'}^l)}{\sqrt{[\nu^2 + \Sigma_{snn}^{A^l}][\nu^2 + \Sigma_{sn'n'}^{A^l}] - (\Sigma_{nn'}^{A^l})^2}}$$

with

$$\mathscr{A}_{srtnn'}^l \triangleq (V_{rn'}^l - \mu_{sn'}^{A^l})^2 [\nu^2 + \Sigma_{snn}^{A^l}] + (V_{tn}^l - \mu_{sn}^{A^l})^2 [\nu^2 + \Sigma_{sn'n'}^{A^l}] +$$
$$2(V_{tn}^l - \mu_{sn}^{A^l})(\mu_{sn'}^{A^l} - V_{rn'}^l)\Sigma_{snn'}^{A^l}$$
$$\mathscr{B}_{snn'}^l \triangleq 2\left( [\nu^2 + \Sigma_{snn}^{A^l}][\nu^2 + \Sigma_{sn'n'}^{A^l}] - (\Sigma_{sn'n}^{A^l})^2 \right).$$

This concludes the calculation of the moments of $F^l$. We can now state how the activation distribution propagates from a layer to the next. The marginal distribution of $A^{l+1}$ is given by

$$\mathrm{Q}(A^{l+1}) = \mathcal{N}(A_{s\star}^{l+1} \mid \widetilde{\mu}_{s\star}^{A^{l+1}}, \widetilde{\Sigma}_{s\star\star}^{A^{l+1}}) \tag{38}$$

with

$$\widetilde{\mu}_{s\star}^{A^{l+1}} = W^{l+1} \widetilde{\mu}_{s\star}^{F^l} \tag{39}$$

$$\widetilde{\Sigma}_{s\star\star}^{A^{l+1}} = W^{l+1} \left( \widetilde{\Sigma}_{s\star\star}^{F^l} + \mathrm{diag}(\sigma^l)^2 \right) (W^{l+1})^T. \tag{40}$$

Thus $\mathrm{Q}(A^L)$ in (31) can be calculated by iterating the application of eqs. (33), (35), (37), (39) and (40) over the layers $l$. To save computational power only the variances can be propagated by assuming that $\widetilde{\Sigma}_{s\star\star}^{F^l}$ is diagonal and therefore ignoring (37).

Now $\L_{\mathrm{pred}}$ can be identified as the expected log-probability of the observations under the marginal distribution $Q(F^L)$ and thus we can expand it as follows,

$$\L_{\mathrm{pred}} = \mathrm{E}_{\mathrm{Q}(F^L)}\left[ \log \mathrm{P}(X^L \mid F^L) \right] \propto \mathrm{E}_{\mathrm{Q}(F^L)}\left[ -S \sum_{n=1}^{N_l} \log \sigma_n^L - \frac{1}{2} \sum_{s=1}^{S} \sum_{n=1}^{N_L} \frac{(X_{sn}^L - F_{sn}^L)^2}{(\sigma_n^L)^2} \right]$$

$$= -S \sum_{n=1}^{N_L} \log \sigma_n^L - \frac{1}{2} \sum_{s=1}^{S} \sum_{n=1}^{N_L} \frac{(X_{sn}^L)^2 - 2 X_{sn}^L \mathrm{E}_{\mathrm{Q}(F^L)}\left[F_{sn}^L\right] + \mathrm{E}_{\mathrm{Q}(F^L)}\left[(F_{sn}^L)^2\right]}{(\sigma_n^L)^2} \tag{41}$$

where $S$ is the number of training samples and $X^L$ are the training targets. The distribution $Q(F^L)$ is of arbitrary form, but only its first and second moments are required to evaluate $\L_{\mathrm{pred}}$. For the first moment we obtain $\mathrm{E}_{\mathrm{Q}(F^L)}\left[F_{sn}^l\right] = \widetilde{\mu}_{sn}^{F^L}$ with $\widetilde{\mu}_{sn}^{F^L}$ given by (33) and the second moment evaluates to

$$\mathrm{E}_{\mathrm{Q}(F^L)}\left[(F_{sn}^L)^2\right] = \mathrm{Var}_{\mathrm{Q}(F^L)}(F_{sn}^L) + \mathrm{E}_{\mathrm{Q}(F^L)}\left[F_{sn}^L\right]^2 = \widetilde{\Sigma}_{snn}^{F^L} + \left(\widetilde{\mu}_{snn}^{F^L}\right)^2$$

$$= 1 - \mathrm{tr}\left[ \left( \widehat{K}_{\star\star n}^{U^L} - \beta_{\star n}^L (\beta_{\star n}^L)^T \right) \Omega_{s\star\star n}^L \right], \tag{42}$$

with $\beta_{\star n}^L \triangleq K(V_{\star n}^L, V_{\star n}^L)^{-1} \widehat{\mu}_{\star n}^{U^L}$ and $\Omega^L$ from (36).

This concludes the calculation of all terms of the variational lower bound (18). The resulting objective is a fully deterministic function of the parameters. Training of the model is performed by maximizing $\L$ w.r.t. to the variational parameters $\widehat{\mu}^{U^l}$, $\widehat{\Sigma}^{U^l}$ and the model parameters $\sigma^l$, $W^l$ and $V^l$. This can be performed using any gradient-descent based algorithm. The necessary derivatives are not derived here and it is assumed that this can be performed automatically using symbolic or automatic differentiation in an appropriate framework.

### 3.3 COMPUTATIONAL AND MODEL COMPLEXITY

The activation function are represented using $2R$ variational parameters per GPN, where $R$ is the number of inducing points and targets. It can be shown that $R = 10$ linearly spaced inducing points are enough to represent the most commonly used activation functions (sigmoid, tanh, soft ReLU) with very high accuracy. The number of required parameters can be reduced by sharing the same activation function within groups of neurons or even across whole layers of neurons. If the inducing points $V^l$ are fixed (for example by equally distributing them in the interval $[-1, 1]$), the kernel matrices $K(V^l, V^l)$ and their inverses can be precomputed since they are constant. The number of parameters and the computational complexity of propagating the means and covariances only depend on $R$ and are therefore *independent* of the number of training samples. Thus, like a conventional neural network, a GPN network can inherently be trained on datasets of unlimited size.

## 4 CONCLUSION

We have presented a non-parametric model based on GPs for learning of activation functions in a multi-layer neural network. We then successively applied variational to make fully Bayesian inference feasible and efficient while keeping its probabilistic nature and providing not only best guess predictions but also confidence estimations in our predictions. Although we employ GPs, our parametric approximation allows our model to scale to datasets of unlimited size like conventional neural networks do.

We have validated networks of Gaussian Process Neurons in a set of experiments, the details of which we submit in a subsequent publication. In those experiments, our model shows to be significantly less prone to overfitting than a traditional feed-forward network of same size, despite having more parameters.

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
