# OpenReview forum: "Gaussian Process Neurons"
_ICLR.cc/2018/Conference — Reject_

### Official Review · AnonReviewer2 · 2017-11-27
**Placing Gaussian process priors on the form of the activation functions in neural nets. The work is interesting but still very preliminary**

**Rating:** 5
**Confidence:** 4

**Review:**

The paper addresses the problem of learning the form of the activation functions in neural networks.  The authors propose to place Gaussian process (GP) priors on the functional form of each activation function (each associated with a hidden layer and unit) in the neural net. This  somehow allows to non-parametrically infer from the data the "shape" of the activation functions needed for a specific problem.  The paper then proposes an inference framework (to approximately marginalize out all GP functions)  based on sparse GP methods that use inducing points and variational inference.  The inducing point approximation used here is very efficient since all GP functions depend on a scalar input (as any activation function!) and therefore by just placing the inducing points in a dense grid gives a fast and accurate representation/compression of all GPs in terms of the inducing function values (denoted by U in the paper).  Of course then inference involves approximating the finite posterior over inducing function values U and the paper make use of the standard Gaussian approximations.

In general I like the idea and I believe that it can lead to a very useful model. However, I have found the current paper quite preliminary and incomplete.  The authors need to address the following:

First (very important): You need to show experimentally how your method compares against regular neural nets (with specific fixed forms for their activation functions such relus etc). At the moment in the last section you mention "We have validated networks of Gaussian Process Neurons in a set of experiments, the details of which we submit in a subsequent publication. In those experiments, our model shows to be significantly less prone to overfitting than a traditional feed-forward network of same size, despite having more parameters." ===>  Well all this needs to be included in the same paper.

Secondly: Discuss the connection with Deep GPs (Damianou and Lawrence 2013). Your method seems to be connected with Deep GPs although there appear to be important differences as well. E.g. you place GPs on the scalar activation functions in an otherwise  heavily parametrized neural network (having interconnection weights between layers) while deep GPs model the full hidden layer mapping as a single GP (which does not require interconnection weights).

Thirdly:  You need to better explain the propagation of uncertainly in section 3.2.2  and the central limit of distribution in section 3.2.1. This is the technical part of your paper which is a non-standard approximation. I will suggest to give a better intuition of the whole idea and move a lot of mathematical details to the appendix.

---

### Official Review · AnonReviewer1 · 2017-11-27
**Intriguing but incomplete work. No experimental validation.**

**Rating:** 4
**Confidence:** 5

**Review:**

In Bayesian neural networks, a deterministic or parametric activation is typically used. In this work, activation functions are considered random functions with a GP prior and are inferred from data.


- Unnecessary complexity

The presentation of the paper is unnecessarily complex. It seems that authors spend extra space creating problems and then solving them. Although some of the derivations in Section 3.2.2 are a bit involved, most of the derivations up to that point (which is already in page 6) follow preexisting literature.

For instance, eq. (3) proposes one model for p(F|X). Eq. (8) proposes a different model for p(F|X), which is an approximation to the previous one. Instead, the second model could have been proposed directly, with the appropriate citation from the literature, since it isn't new. Eq. (13) is introduced as a "solution" to a non-existent problem, because the virtual observations are drawn from the same prior as the real ones, so it is not that we are "coming up" with a convenient GP prior that turns out to produce a computationally tractable solution, we are just using the prior on the observations consistently.

In general, the authors seem to use "approximately equal" and "equal" interchangeably, which is incorrect. There should be a single definition for p(F|X). And there should be a single definition for L_pred. The expression for L_pred given in eq. (20) (exact) and eq. (41) (approximate) do not match and yet both are connected with an equality (or proportionality), which they shouldn't.

Q(A) is sometimes taken to mean the true posterior (i.e., eq. (31)), sometimes a Gaussian approximation (i.e., eq (32) inside the integral), and both are used interchangeably.


- Incorrect references to the literature

Page 3: "using virtual observations (originally proposed by Quiñonero-Candela & Rasmussen (2005) for sparse approximations of GPs)"

The authors are citing as the origin of virtual observations a survey paper on the topic. Of course, that survey paper correctly attributes the origin to [1].

Page 4: "we apply the technique of variational inference Wainwright et al. (2008)".

How can variational inference be attributed to (again) a survey paper on the topic from 2008, when for instance [2] appeared in 2003?


- Correctness of the approach

Can the authors guarantee that the variational bound that they are introducing (as defined in eqs. (19) and (41)) is actually a variational bound? It seems to me that the approximations made to Q(A) to propagate the uncertainty are breaking the bounding guarantee. If it is no longer a lower bound, what is the rationale behind maximizing it?

The mathematical basis for this paper is actually introduced in [3] and a single-layer version of the current model is developed in [4]. However, in [4] the authors manage to avoid the additional Q(A) approximation that breaks the variational bound. The authors should contrast their approach with [4] and discuss if and why that additional central limit theorem application is necessary.


- No experiments

The use of a non-parametric definition for the activation function should be contrasted with the use of a parametric one. With enough data, both might produce similar results. And the parameter sharing in the parametric one might actually be beneficial. With no experiments at all showing the benefit of this proposal, this paper cannot be considered complete.


- Minor errors:

Eq. (4), for consistency, should use the identity matrix for the covariance matrix definition.
Eq. (10) uses subscript d where it should be using subscript n
Eq. (17) includes p(X^L|F^L) in the definition of Q(...), but it shouldn't. That was particularly misleading, since if we take eq. (17) to be correct (which I did at first), then p(X^L|F^L) cancels out and should not appear in eq. (20).
Eq. (23) uses Q(F|A) to mean the same as P(F|A) as far as I understand. Then why use Q?


- References

[1] Edward Snelson and Zoubin Ghahramani. Sparse Gaussian processes using pseudo-inputs.
[2] Beal, M.J. Variational Algorithms for Approximate Bayesian Inference.
[3] M.K. Titsias and N.D. Lawrence. Bayesian Gaussian process latent variable model.
[4] M. Lázaro-Gredilla. Bayesian warped Gaussian processes.

---

> ### Public Comment · (anonymous) · 2017-12-05
> **Strongly agree**
>
> I agree with this reviewer. Much of the mathematical derivation has been worked out before, even much of the uncertainty propagation part. I would add that [1] reviews many of the papers relying on these derivations.
>
> While the paper proposes an interesting model, I believe the paper can't really be accepted without any experimental verification.
>
> [1] http://jmlr.org/papers/volume17/damianou16a/damianou16a.pdf

---

### Official Review · AnonReviewer3 · 2017-11-28
**GPN Review**

**Rating:** 7
**Confidence:** 2

**Review:**

This paper investigates probabilistic activation functions that can be structured in a manner similar to traditional neural networks whilst deriving an efficient implementation and training regime that allows them to scale to arbitrarily sized datasets.

The extension of Gaussian Processes to Gaussian Process Neurons is reasonably straight forward, with the crux of the paper being the path taken to extend GPNs from intractable to tractable.
The first step, virtual observations, are used to provide stand ins for inputs and outputs of the GPN.
These are temporary and are later made redundant.
To avoid the intractable marginalization over latent variables, the paper applies variational inference to approximate the posterior within the context of given training data.
Overall the process by which GPNs are made tractable to train leverages many recent and not so recent techniques.

The resulting model is theoretically scalable to arbitrary datasets as the total model parameters are independent of the number of training samples.
It is unfortunate but understandable that the GPN model experiments are confined to another paper.

---

### Decision · Program_Chairs · 2018-01-29
**ICLR 2018 Conference Acceptance Decision**

**Decision:**

Reject

**Comment:**

The authors propose the use of Gaussian processes as the prior over activation functions in deep neural networks.  This is a purely mathematical paper in which the authors derive an efficient and scalable approach to their problem.  The idea of having flexible distributions over activation functions is interesting and possibly impactful.  One reviewer recommended acceptance with low confidence.  The other two found the idea interesting and compelling but confidently recommended rejection.  These reviewers are concerned that the paper is unnecessarily complex in terms of the mathematical exposition and that it repeats existing derivations without citation.  It is very important that the authors acknowledge existing literature for mathematical derivations.  Furthermore, the reviewers question the correctness of some of the statements (e.g. is the variational bound preserved?).  These reviewers agreed that the paper is incomplete without any empirical validation.

Pros:
- A compelling and promising idea
- The approach seems to be scalable and highly plausible

Cons:
- No experiments
- Significant issues with citing of related work
- Significant questions about the novelty of the mathematical work